# Predictors and outcomes of recognition of intellectual disability for adults during hospital admissions: A retrospective data linkage study in NSW, Australia

Adrian Raymond Walker[1], Julian Norman Trollor[1]*, Tony Florio[2], Preeyaporn Srasuebkul[1]

1 The Department of Developmental Disability Neuropsychiatry, The University of New South Wales Sydney, New South Wales, Australia, 2 Australian Catholic University, New South Wales, Australia

* j.trollor@unsw.edu.au

## Abstract

Adults with intellectual disability have high health care needs. Despite frequent contact with health services, they often receive inadequate health care. One method to improve health care delivery is reasonable adjustments, that is, the adaptation of health care delivery such that barriers to participation are removed for the person with disability. A starting point for the provision of reasonable adjustments is recognition of intellectual disability during the health care contact. To determine rates and predictors of the recognition of intellectual disability during hospital admissions, and its impact on admission metrics, we examined a population of adults with intellectual disability identified from disability services datasets from New South Wales, Australia between 2005 and 2014. Recognition of intellectual disability was determined by the recording of an International Classification of Diseases 10th revision (ICD-10) diagnostic code for intellectual disability during a given hospital admission. We examined how recognition of intellectual disability related to length of hospital episodes. We found an overall low rate of recognition of intellectual disability (23.79%) across all hospital episodes, with the proportion of hospital episodes recognising intellectual disability decreasing from 2005–2015. Admissions for adults with complex health profiles (e.g., those with many comorbidities, those with Autism Spectrum Disorder, and those admitted for urgent treatment) were more likely to recognise intellectual disability, but admissions for adults with complexity in other domains (i.e., for those in custody, or those with drug and alcohol disorders) were less likely to recognise intellectual disability. Recognition of intellectual disability was associated with longer episodes of care, possibly indicating the greater provision of reasonable adjustments. To improve the recognition of intellectual disability for adults during health service contacts, we advocate for the implementation of targeted initiatives (such as a nationwide disability flag to be included in health service records) to improve the provision of reasonable adjustments.

**Data Availability Statement:** Our datasets cannot be shared publicly due to the data usage agreement between the Department of

Developmental Disability Neuropsychiatry, The University of New South Wales Sydney, and the data custodians who provide access to our data. Researchers who meet the criteria for obtaining access to confidential data who wish to access the data should enquire to the NSW Population and Health Services Research Ethics Committee (cinsw-ethics@health.nsw.gov.au) and quote the project and sub-study reference number (2013/02/446, 2019UMB0209).

**Funding:** This study was funded by a National Health and Medical Research Council Australia (https://www.nhmrc.gov.au/) funded Partnerships for Better Health grant (APP1056128; Title: Improving the Mental Health Outcomes of People with an Intellectual Disability; Awarded to JT) and a National Health and Medical Research Council Australia Project grant (APP1123033; Title: Understanding health service system needs for people with intellectual disability; Awarded to JT). JT is funded as the Chair of Intellectual Disability Mental Health by the Mental Health Branch of the NSW Ministry of Health. The funders had no role in study design, data collection and analysis, decision to publish, or preparation of the manuscript.

**Competing interests:** The authors have declared that no competing interests exist.

## Introduction

People with intellectual disability have high health needs that require frequent use of acute health care services such as emergency departments and hospitalisations [1, 2]. People with intellectual disability often experience poor quality of care and inefficiency of care in these settings [3, 4], which contribute to reattendance at emergency departments, and readmission within close proximity to discharge [5, 6]. Failures of acute care are part of a broader health gap experienced by people with intellectual disability, including substantially reduced life expectancy [7] and a very high proportion of deaths from potentially avoidable causes [8, 9]. In Australia, this health gap has been characterised as systematic neglect by the Australian Royal Commission into Violence, Abuse, Neglect and Exploitation of People with Disability [3]. Developing solutions to this problem is a matter of urgency for this population.

One way to improve the experience of people with intellectual disability as they interact with the health care system is to ensure reasonable adjustments are made during health service contacts [10, 11]. Reasonable adjustments are defined as removing barriers to services that may affect people with disability, such as changing the way services are delivered, altering policies or procedures as appropriate, and providing staff with the proper training to meet the service needs of people with disability [10, 12], and are mandated during health service contacts in the United Kingdom for people with intellectual disability [13]. A recent metareview into reasonable adjustments in the health care system for people with intellectual disability suggested that changes like preadmission hospital visits, extended consultation times, and access to speciality intellectual disability nurses can substantially improve the quality of care received during an inpatient hospital episode by people with intellectual disability [10, 11, 14–16]. However, reasonable adjustments cannot be reliably employed within health service settings unless intellectual disability is identified.

To improve the provision of reasonable adjustments, the United Kingdom's National Health Service has recently introduced a nationwide "Reasonable Adjustments Flag" to indicate where patient may need reasonable adjustments during a service contact [17]. Such a flag assists identification of the disability, and triggers the need to ask about and document the adjustments required by the person during their contact with the health service. However, research suggests that intellectual disability is not consistently recognised during receipt of health and human services, and that recognition varies according to the sector with which the person has contact, creating an impediment to the uniform application of reasonable adjustments [11, 18–21]. Though previous research has examined factors that affect the recognition of intellectual disability in children [20], the factors that determine recognition of intellectual disability for adults during inpatient episodes (and therefore provision of reasonable adjustments) and the impact of this identification on the episode of care have not been identified.

Using a data linkage population from New South Wales (NSW) Australia [22], we aimed to investigate what demographic and health factors affected the recognition of intellectual disability during contact with inpatient hospital services for adults. Furthermore, as no studies to date have leveraged large-linked datasets to examine how correctly identifying intellectual disability can affect the trajectory of an inpatient episode, we also aimed to investigate how the recognition of intellectual disability during an inpatient episode affected the length of that inpatient episode in our linked population. In doing so, we hoped to expand our understanding of the impacts of intellectual disability labelling during contacts with health services, and provide support for improved systems for recognising intellectual disability in health care systems worldwide.

## Methods

### Datasets, record linkage

This study was a retrospective cohort study, conducted as a substudy within a bigger data linkage infrastructure project described in Reppermund et al. [22]. All data used in this study were collected into administrative datasets during the interaction of members of the cohort with administrative services (e.g., hospital or disability services). No participant data were gained through direct interaction with members of the cohort, such as via structured interviews. Linkage was performed by the NSW Centre for Health Record Linkage using best practice methods.

We used three datasets to define a population of adults known to have an intellectual disability within a disability related dataset: the New South Wales (NSW) disability services: the Disability Minimum Services Data Set (DS-MDS), which contains information on all people who received a disability service in NSW between 1 Jul 2005 to 30 Jun 2015; the NSW Public Guardian Data Set (PG), which contains information on all people who accessed Public Guardian services for decision-making assistance relating to health and lifestyle in NSW between 1 Jan 1994 and 30 Apr 2016; and the State-wide Disability Services Data Set (SDS), which contains information on those who received disability services while in custody between 1 Jan 2001 to 31 May 2016.

We obtained information on hospital admissions from the Admitted Patients Data Collection (APDC), which records information about hospital episodes that occurred in NSW between 1 Jul 2001 to 30 Mar 2016, including a deidentified personal identifier, start and end dates and times, hospital location, principal diagnosis and up to 50 additional diagnoses, and some personal information such as a person's sex, age, and current area of residence. Only hospital admissions that occurred after an individual had been identified in one or more of the disability datasets were included in this study. Information about whether an admission occurred during an episode of custody was also available to us through linkage to the Offender Integrated Management System (OIMS), and from which records were available for the study duration. We determined demographic data as per the linkage method described in Reppermund et al. [22].

### Study population

The study population comprised of people with intellectual disability of ages 18 and over known to NSW disability services who appeared in the DS-MDS, PG, or SDS between 1 Jul 2005 and 30 Jun 2014, with their first appearance being the index date for the study (noting that this allowed some people to have an index date prior to the date they turned 18, as long as they turned 18 between 1 Jul 2005 and 30 Jun 2014). We excluded from the study population people with intellectual disability who did not have any hospital episodes after the index date in our study period. Persons younger than 18 years of age were excluded.

### Follow up period

The follow up period for each person started from either their index date, the date of their first appearance in one of these datasets (if they were over 18), or the date they turned 18 (if they had appeared in one of these datasets at an earlier date), whichever occurred last. Their follow up ended at 30 Jun 2015 or their date of death, whichever occurred first.

### Outcome variables

We measured our outcome variables at an episode of care level. The episode of care is the period of admitted patient care between a formal (cessation of a stay in hospital) or statistical

(cessation of an episode occurring within a hospital stay, which may be followed by another episode) separation, characterised by only one care type [23–25]. That is, an episode of care can be thought of as a discrete contact with the hospital inpatient system where a person receives a particular type of care (e.g., mental health care, dialysis, rehabilitation, etc). We included all hospital episodes that occurred for an individual within the follow-up period when determining our outcome variables. Individuals were considered as having the first outcome variable, intellectual disability recognition, if intellectual disability was recorded in a hospital episode as a principal or additional diagnosis. We used the International Classification of Diseases 10th Edition (ICD-10) codes recorded for each hospital episode to identify the presence or absence of intellectual disability in a hospital episode (F7, F84.2, Q90, Q91, Q93, Q95-99. Q86, Q87, Q89.8, P04.3) [22].

The second outcome variable was the length of stay in days for a given hospital episode, obtained by the duration in days between episode start and end as recorded in the APDC. Episodes that started and finished within the same day were considered to have a length of stay one day. Episodes that were recorded as occurring within another episode (i.e., 'nested' episodes), were included as it could not be determined that these stays did not constitute discrete service contacts.

A description of the steps taken to form the study population can be seen in Fig 1.

## Statistical analysis

We established the sociodemographic profile of the people who had valid episodes during the study period, the characteristics of valid episodes during the study period, and the raw proportion of hospital episodes where intellectual disability was recognised by financial year. All

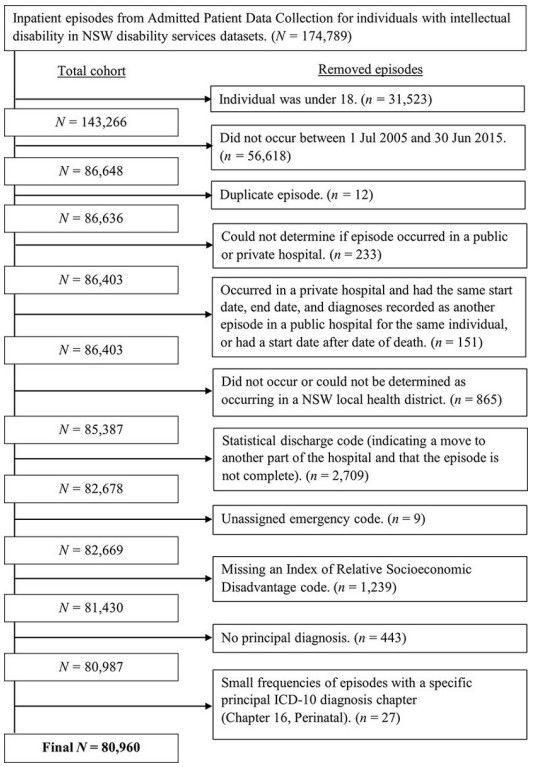

**Fig 1. Description of the steps taken for study population formation.**

demographic information for this description was obtained using the linkage method described Reppermund et al. [22].

Intellectual disability recognition

We conducted a multi-level logistic regression in Stata 17 to find predictors of intellectual disability recognition [26]. We used multi-level regression as people often present to hospital more than once in the study period. By including a random effect of person into our model, we assume that an individual's hospital episodes will share some variance (i.e., we assume two episodes from one individual will be more similar than two episodes from two different individuals), allowing us to better control for the overall variance in the model. We tested this assumption with a likelihood-ratio test of fit against a standard logistic regression without a random effect of person.

Fixed effects for the multi-level regression can be seen in Table 1. From the variables available in our datasets, we selected fixed effects for our models through a combination of expert medical knowledge about intellectual disability (JT), and internal discussion between the authors based on previous literature [9, 20, 27]. We included all variables selected in our analysis of intellectual disability recognition as fixed effects (besides intellectual disability recognition, which was the outcome variable for this model). Most fixed effects included were at the

**Table 1. Fixed effects included in multi-level regression modelling.**

| Fixed Effect | Type | Level | Definition |
|---|---|---|---|
| Sex | Binary | Person | Sex of the individual as given by demographic data. |
| Financial year | Categorical | Episode | Financial year in which the episode occurred. |
| Age | Continuous | Episode | Age when the episode started. |
| Years after recognition | Continuous | Episode | Years since the individual was first recorded in the DS-MDS, PG, or SDS, calculated from the episode end date. |
| Local Health District location | Categorical | Episode | Local Health District in which an episode occurred, defined as per New South Wales Health categories (including St Vincent's in the Specialty category) [28]. |
| Remoteness | Categorical | Episode | An individual's remoteness category, as determined by Statistical Area 2 codes given by the Australian Bureau of Statistics [29]. In cases where one code contributed to multiple remoteness areas, the remoteness with the highest percentage was used. |
| Aboriginal or Torres Strait Islander | Binary | Person | Whether an individual was identified as Aboriginal or Torres Strait Islander as given by demographic data (output withheld for ethical reasons). |
| Index of relative socioeconomic disadvantage (IRSD) quintile | Categorical | Episode | Measure of disadvantage recorded during an episode, determined by Statistical Area 2 codes given by the Australian Bureau of Statistics [30]. |
| In custody at time of episode | Binary | Episode | Whether an episode occurred while the individual was in custody, determined by the OIMS dataset. |
| Drug and Alcohol episode | Binary | Episode | Whether the episode had a record of drug or alcohol related ICD-10 diagnosis codes (either principal or additional diagnoses), using the definition from the Australian Institute of Health and Welfare [31]. |
| Presence of Autism Spectrum Disorder | Binary | Person | Whether an individual was recorded to have Autism Spectrum Disorder as given by demographic data. |
| Urgent admission | Binary | Episode | Whether the episode had a flag indicating the individual required attention within 24 hours. |
| Summed Elix-Hauser comorbidities | Continuous | Episode | Number of Elix-Hauser comorbidity diagnoses recorded in an episode (either as principal or additional diagnoses) [32, 33]. |
| Hospital type | Categorical | Episode | Whether the episode was recorded as occurring in a public or private hospital. |
| Diagnosis chapter | Categorical | Episode | ICD-10 diagnosis chapter to which the principal diagnosis for that episode belonged [34]. We used Chapter 5 Mental disorders as a reference group. |
| Intellectual disability recognition (length of stay model only) | Binary | Episode | Whether an episode had an ICD-10 code recognising intellectual disability. |

DS-MDS = Disability Services Minimum Data Set; PG = Public Guardian; SDS = State-wide Disability Services Data Set; OIMS = Offender Integrated Management System; ICD-10 = International Classification of Diseases 10th edition

All variables are from the Admitted Patients Data Collection, unless otherwise specified.

episode level (i.e., from data obtained directly from the records in that episode), with some variables (including age, Aboriginal and Torres Strait Islander Status, presence of Autism Spectrum Disorder) determined for each person based on the linkage method described in Reppermund et al. [22]. We used ICD-10 chapter 5 as the reference level for the principal diagnosis chapter variable as most ICD-10 codes for intellectual disability are in this chapter (and may include cases where the principal diagnosis was intellectual disability). We included a random effect of person identifier into the model. We did not include any ICD10 chapters where there were no records.

We estimated the impact of effects in the model using the estimated likelihood, after fitting the regression model. The estimated likelihood represents the estimated average likelihood of intellectual disability recognition when a specific fixed effect changes and other covariates are assumed to be as recorded. For example, the estimated average likelihood in 2005–2006 financial year is calculated as if all episode in the study population occurred in 2005–2006 and other variables remain as they are recorded. The estimated likelihood was calculated for each level of each categorical fixed effect, and at selected point estimates for each continuous fixed effect (age, years after recognition, and summed Elix-Hauser comorbidities).

**Length of stay.**   We conducted a multi-level Poisson regression to determine predictors of length of stay for an episode. We used the same fixed and random effects as the above model with an addition of a fixed effect of whether the episode recognised intellectual disability. We tested whether the multi-level Poisson regression was necessary with a likelihood ratio test of fit against a standard Poisson regression without a random effect of person. Finally, we calculated the estimated length of stay for each level of each variable (in the same fashion as the estimated likelihood in the model of intellectual disability recognition above).

**Subset analysis.**   To ensure that our results were not biased by including people who were identified in the DS-MDS, PG, or SDS as having intellectual disability before they turned 18, we conducted two subset analyses (one for intellectual disability recognition, and one for length of stay) restricting the study population to only individuals who were over 18 at their index date (i.e., when they first appeared in the DS-MDS, PG, or SDS). The results of these regressions are reported alongside our other analyses for comparison.

## Ethics approval

The study was approved by the NSW Population and Health Services Research Ethics Committee (HREC/13/CIPHS/7; Cancer Institute NSW reference: 2013/02/446 and Sub-study Reference number: 2019UMB0209). This approval included a waiver of informed consent.

## Results

### Sociodemographics and episode characteristics

Table 2 shows the sociodemographic profile of the study population, and Table 3 shows the characteristics of hospital episodes for our study population. Overall, there were 12,593 individuals with 80,960 hospital episodes in the study period. Over half of the study population were male (58.1%), 65.7% lived in major cities, and 49.9% lived in disadvantaged areas. Overall, intellectual disability was recognised in 19,261 (23.79%) of all episodes. Most episodes (51.7%) occurred after the start of the 2011 financial year in metropolitan local health districts (63.3%). Around half of all admissions were urgent episodes (50.5%). Most episodes occurred in a public hospital (92.2%), and ICD-10 chapter 21 (Factors influencing health contact) accounted for the largest percentage of episodes (23.8%), noting that this chapter includes codes that often require regular hospital admissions (such as dialysis, ICD-10 code Z49, and rehabilitation, ICD-10 code Z50). Fig 2 shows the raw proportion of hospital episodes where

**Table 2. Sociodemographic profile of the study population.**

| Variable | Frequency (% of people) |
|---|---|
| **Number of people** | **12,593** |
| **Sex** | |
| Male | 7,317 (58.1%) |
| Female | 5,276 (41.9%) |
| **Age at index** | |
| Median | 32.8 |
| Inter-quartile range | 20.0–47.1 |
| **Remoteness of area of residence** | |
| Major Cities of Australia | 8,275 (65.7%) |
| Inner Regional Australia | 3,022 (24.0%) |
| Outer Regional Australia and Beyond | 1,296 (10.3%) |
| **Index of Relative Socio-economic Disadvantage Quintile** | |
| 1(Most disadvantaged) | 2,644 (21.0%) |
| 2 | 3,640 (28.9%) |
| 3 | 2,633 (20.9% |
| 4 | 1,805 (14.3%) |
| 5 (Least disadvantaged) | 1,609 (12.8%) |
| Missing | 262 (2.1%) |
| **Indigenous status** | 1,342 (10.7%) |
| **Autism Spectrum Disorder** | 747 (5.9%) |

intellectual disability was recognised by financial year. Initially, intellectual disability was recognised in approximately 35% of hospital episodes, but this decreased in 2008–2009 to around 25% of episodes, and continued to decrease to around 20% of episodes by 2014–2015.

## Diagnostic recognition of intellectual disability

Table 4 shows the results of the multi-level logistic regression predicting intellectual disability recognition within a hospital episode (with a subset analysis restricting the study population to only those with an index date after they turn 18 shown in Table 5). The multi-level logistic regression with a random effect of person provided a significantly better fit than an equivalent logistic regression without a random effect of person ($\bar{\chi}^2(1) > 10,000$, $p < .001$), indicating that accounting for individual level variance improved the fit of our model. Overall, women with intellectual disability were more likely to be recognised as having an intellectual disability within a hospital episode than men (odds ratio (OR): 1.11, 95% confidence interval (CI): 1.02–1.21), and the older someone was at the time of their episode the more likely they were to be recognised as having an intellectual disability (OR: 1.01, 95%CI: 1.01–1.01). Recognition of intellectual disability decreased across financial years ($\chi^2(9) = 1033.75$, $p < 0.001$), but increased the more years a person had been known to disability services (OR: 1.25, 95%CI: 1.22–1.27). Episodes were more likely to recognise intellectual disability if they occurred in rural local health districts (OR: 1.51, 95%CI: 1.37–1.67), but less likely if they occurred in speciality local health districts (OR: 0.37, 95%CI: 0.29–0.46) compared to episodes that occurred in metropolitan local health districts. However, the more remote the residence of the person presenting in an episode, the less likely that episode was to recognise intellectual disability ($\chi^2(2) = 78.48$, $p < 0.001$). Autistic people were more likely to be recognised as having an intellectual disability (OR: 1.62, 95%CI: 1.44–2.83). Episodes where a person was in custody at the time of the episode (OR: 0.09, 95%CI: 0.05–0.15), or had drug and alcohol related diagnoses (OR: 0.68,

**Table 3. Characteristics of hospital episodes.**

| Variable | Frequency (% of episodes) |
|---|:---:|
| **Number of episodes** | **80,960** |
| **Intellectual disability recognition** | |
| No | 61,699 (76.2%) |
| Yes | 19,261 (23.8%) |
| **Financial year** | |
| 2005–2006 | 4,321 (5.3%) |
| 2006–2007 | 5,232 (6.5%) |
| 2007–2008 | 5,718 (7.1%) |
| 2008–2009 | 6,910 (8.5%) |
| 2009–2010 | 7,993 (9.9%) |
| 2010–2011 | 8,892 (11%) |
| 2011–2012 | 9,723 (12%) |
| 2012–2013 | 10,674 (13.2%) |
| 2013–2014 | 10,974 (13.6%) |
| 2014–2015 | 10,523 (13%) |
| **Age at episode (median (IQR))** | |
| Median | 39.2 |
| Inter-quartile range | 26.8–51.7 |
| **Local Health District location** | |
| Metropolitan | 51,236 (63.3%) |
| Rural | 27,798 (34.3%) |
| Speciality | 1,926 (2.4%) |
| **Remoteness** | |
| Major cities | 57,055 (70.5%) |
| Inner regional | 18,187 (22.5%) |
| Outer regional and beyond | 5,718 (7.1%) |
| **IRSD quintile** | |
| 1 (Most disadvantaged) | 25,535 (31.5%) |
| 2 | 17,337 (21.4%) |
| 3 | 17,335 (21.4%) |
| 4 | 9,668 (11.9%) |
| 5 (Least disadvantaged) | 11,085 (13.7%) |
| **In custody at time of episode** | |
| No | 79,786 (98.5%) |
| Yes | 1,174 (1.5%) |
| **Drug and alcohol disorder episode** | |
| No | 76,181 (94.1%) |
| Yes | 4,779 (5.9%) |
| **Urgent admission** | |
| No | 40,042 (49.5%) |
| Yes | 40,918 (50.5%) |
| **Summed Elix-Hauser comorbidities** | |
| 0 | 35,279 (43.6%) |
| 1–2 | 42,243 (52.2%) |
| 3+ | 3,438 (4.2%) |
| **Hospital type** | |
| Public | 74,619 (92.2%) |

(*Continued*)

**Table 3.** (Continued)

| Variable | Frequency (% of episodes) |
|---|---|
| Private | 6,341 (7.8%) |
| **Diagnosis chapter** | |
| 1. Infectious and parasitic | 1,421 (1.8%) |
| 2. Neoplasms | 1,915 (2.4%) |
| 3. Blood and blood forming | 877 (1.1%) |
| 4. Endocrine | 1,471 (1.8%) |
| 5. Mental and behavioural | 9,835 (12.1%) |
| 6. Nervous | 5,804 (7.2%) |
| 7. Eye and adnexa | 1,325 (1.6%) |
| 8. Ear and mastoid | 387 (0.5%) |
| 9. Circulatory | 1,730 (2.1%) |
| 10. Respiratory | 4,964 (6.1%) |
| 11. Digestive | 10,190 (12.6%) |
| 12. Skin and subcutaneous | 1,992 (2.5%) |
| 13. Musculoskeletal | 1,782 (2.2%) |
| 14. Genitourinary | 2,853 (3.5%) |
| 15. Pregnancy and the puerperium | 1,081 (1.3%) |
| 17. Congenital and chromosomal | 436 (0.5%) |
| 18. Abnormal signs and symptoms | 6,922 (8.5%) |
| 19. Injury and poisoning | 6,743 (8.3%) |
| 21. Factors influencing contact | 19,232 (23.8%) |

95%CI: 0.60–0.76), were less likely to recognise intellectual disability. Episodes that were urgent (OR: 1.37, 95%CI: 1.28–1.46) were more likely to recognise intellectual disability. A higher number of Elix-Hauser comorbidities within an episode was associated with a greater likelihood of recognition of intellectual disability (OR: 1.49, 95%CI: 1.45–1.54). Episodes that occurred in private hospitals were less likely to recognise intellectual disability (OR: 0.26, 95% CI: 0.23–0.30). The average estimated likelihoods showed episodes with a principal diagnosis in ICD-10 chapter 17 (Congenital and chromosomal) had the highest likelihood of recognising intellectual disability (marginal likelihood = 0.43, 95%CI: 0.38–0.47), while episodes where the primary diagnosis was in ICD-10 chapter 3 (Blood and blood forming) had the lowest likelihood of intellectual disability recognition (marginal likelihood = 0.20, 95%CI: 0.07–0.23). We did not observe an effect of socioeconomic disadvantage on predicting recognition of intellectual disability ($\chi^2(4) = 7.42$, $p = 0.115$). When considering the subset analysis with only people with index dates after they turned 18, all results were similar in their direction and significance, with the exception that there was no significant effect of sex (OR: 1.05, 95%CI: 0.96–1.16).

## Length of hospital episode

Table 6 shows the results of the multi-level logistic regression predicting intellectual disability recognition within a hospital episode (with a sensitivity analysis restricting the study population to only those with an index date after they turn 18 shown in Table 7). The multi-level Poisson regression with a random effect of person provided a significantly better fit than an equivalent Poison regression without a random effect of person ($\bar{\chi}^2(1) > 10,000$, $p < .001$), indicating that accounting for individual level variance improved the fit of our model. Notably, episodes where intellectual disability was recognised were substantially longer than those where intellectual disability was not recognised (IRR: 1.57, 95%CI: 1.56–1.59). The subset

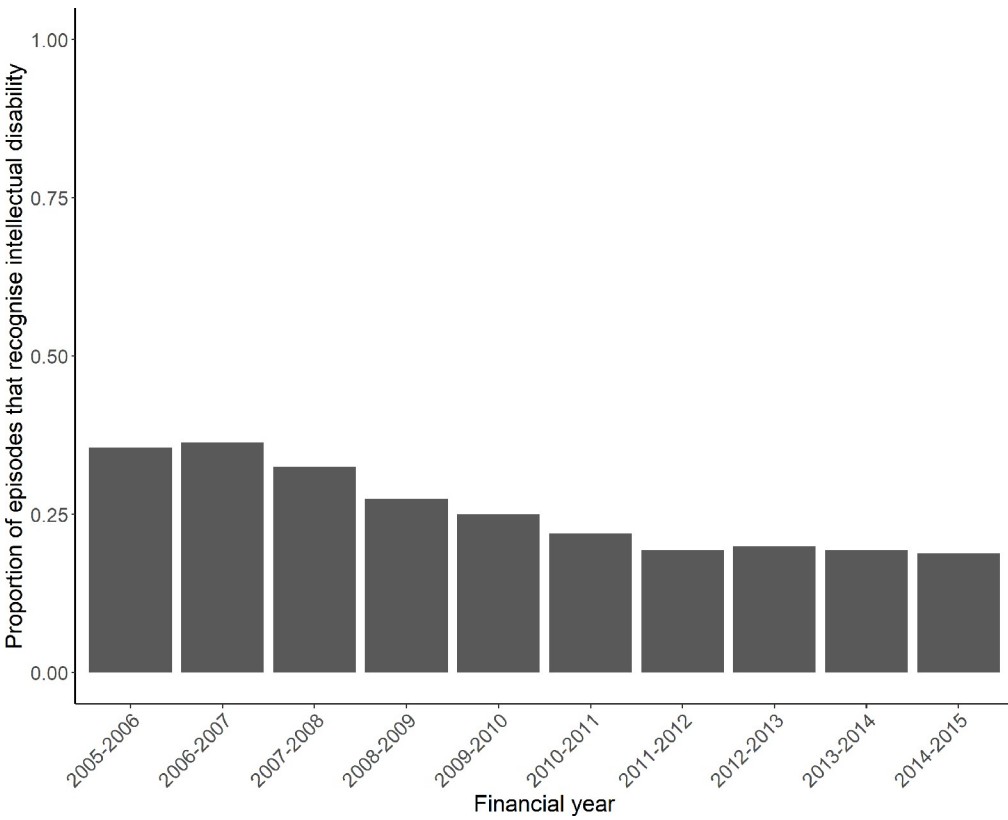

**Fig 2. Raw proportion of episodes that recognise intellectual disability (by financial year).**

analysis with only people with index dates after they turned 18 showed a similar effect of intellectual disability recognition on length of stay (IRR: 1.59, 95%CI: 1.57–1.60)

## Discussion

The current study aimed to investigate the factors associated with recognition of intellectual disability during an inpatient episode in adults, and how this recognition affects the length of inpatient episodes in a population of people with intellectual disability from NSW, Australia. We ran two mixed-effect regressions, one predicting the presence of an intellectual disability diagnosis during an inpatient episode, and one predicting the length of stay of an inpatient episode for adults with intellectual disability. We found that recognition of intellectual disability during an episode was more likely if the person was a woman, was older, had been known to disability services for a longer period of time, was Autistic, or had a high number of Elix-Hauser comorbidities, and was also more likely if the episode occurred in a rural local health district, or was an urgent admission. Recognition of intellectual disability during an episode was less likely if the episode occurred in a specialty local health network, occurred in a private hospital, occurred while the person was in custody, had a drug or alcohol related code recorded, or the person lived outside a major city. For length of stay, we found that the recognition of intellectual disability in an inpatient episode predicted a longer length of stay, after controlling for other demographic and health variables. Taken together, our findings suggest that adults with more complex health needs are more likely to be recognised as having intellectual disability, but adults with complexity across other domains (such as drug and alcohol problems, and contacts with the justice system) are less likely to be recognised as having an intellectual disability.

**Table 4. Predictors of intellectual disability recognition in a hospital episode (n = 12,593).**

| Variable | Odds Ratio (95% CI) | SE | p value | Marginal likelihood (95% CI) |
|---|---|---|---|---|
| **Sex** | | | | |
| Male | *Reference* | . . . | . . . | 0.29 (0.28, 0.30) |
| Female | 1.11 (1.02, 1.21) | 0.05 | 0.020 | 0.30 (0.29, 0.31) |
| **Age (years)** | 1.01 (1.01, 1.01) | < .01 | <0.001 | . . . |
| 20 | . . . | . . . | . . . | 0.27 (0.26, 0.28) |
| 30 | . . . | . . . | . . . | 0.28 (0.27, 0.29) |
| 40 | . . . | . . . | . . . | 0.29 (0.29, 0.30) |
| 50 | . . . | . . . | . . . | 0.31 (0.30, 0.32) |
| 60 | . . . | . . . | . . . | 0.32 (0.31, 0.33) |
| 70 | . . . | . . . | . . . | 0.34 (0.33, 0.35) |
| 80 | . . . | . . . | . . . | 0.36 (0.34, 0.37) |
| **Financial year** | . . . | . . . | <0.001 | . . . |
| 2005–2006 | *Reference* | . . . | . . . | 0.55 (0.53, 0.57) |
| 2006–2007 | 0.90 (0.79, 1.01) | 0.06 | 0.084 | 0.53 (0.51, 0.55) |
| 2007–2008 | 0.62 (0.54, 0.70) | 0.04 | <0.001 | 0.48 (0.46, 0.49) |
| 2008–2009 | 0.36 (0.31, 0.41) | 0.02 | <0.001 | 0.39 (0.38, 0.41) |
| 2009–2010 | 0.26 (0.22, 0.30) | 0.02 | <0.001 | 0.35 (0.34, 0.36) |
| 2010–2011 | 0.17 (0.14, 0.20) | 0.01 | <0.001 | 0.29 (0.28, 0.30) |
| 2011–2012 | 0.11 (0.10, 0.13) | 0.01 | <0.001 | 0.24 (0.23, 0.25) |
| 2012–2013 | 0.10 (0.08, 0.12) | 0.01 | <0.001 | 0.23 (0.22, 0.24) |
| 2013–2014 | 0.08 (0.06, 0.09) | 0.01 | <0.001 | 0.20 (0.19, 0.21) |
| 2014–2015 | 0.05 (0.04, 0.07) | 0.01 | <0.001 | 0.16 (0.15, 0.17) |
| **Years after recognition** | 1.25 (1.22, 1.27) | 0.01 | <0.001 | . . . |
| 0 | . . . | . . . | . . . | 0.19 (0.18, 0.20) |
| 2 | . . . | . . . | . . . | 0.24 (0.23, 0.25) |
| 4 | . . . | . . . | . . . | 0.29 (0.29, 0.30) |
| 6 | . . . | . . . | . . . | 0.35 (0.34, 0.36) |
| 8 | . . . | . . . | . . . | 0.41 (0.40, 0.42) |
| 10 | . . . | . . . | . . . | 0.47 (0.45, 0.49) |
| **Local Health District location** | . . . | . . . | <0.001 | . . . |
| Metropolitan | *Reference* | . . . | . . . | 0.28 (0.27, 0.29) |
| Rural | 1.51 (1.37, 1.67) | 0.08 | <0.001 | 0.33 (0.32, 0.34) |
| Specialty | 0.37 (0.29, 0.46) | 0.04 | <0.001 | 0.17 (0.15, 0.19) |
| **Remoteness** | . . . | . . . | <0.001 | . . . |
| Major cities | *Reference* | . . . | . . . | 0.31 (0.31, 0.32) |
| Inner regional | 0.68 (0.61, 0.76) | 0.04 | <0.001 | 0.26 (0.25, 0.27) |
| Outer regional and beyond | 0.50 (0.42, 0.59) | 0.04 | <0.001 | 0.23 (0.21, 0.25) |
| **IRSD quintile** | . . . | . . . | 0.115 | . . . |
| 1 (Most disadvantaged) | *Reference* | . . . | . . . | 0.29 (0.28, 0.30) |
| 2 | 1.09 (0.99, 1.19) | 0.05 | 0.087 | 0.30 (0.29, 0.31) |
| 3 | 1.00 (0.91, 1.11) | 0.05 | 0.943 | 0.29 (0.28, 0.30) |
| 4 | 1.13 (1.00, 1.27) | 0.07 | 0.048 | 0.31 (0.29, 0.32) |
| 5 (Least disadvantaged) | 1.01 (0.89, 1.14) | 0.06 | 0.925 | 0.29 (0.28, 0.31) |
| **In custody at time of episode** | | | | |
| No | *Reference* | . . . | . . . | 0.30 (0.29, 0.30) |
| Yes | 0.09 (0.05, 0.15) | 0.02 | <0.001 | 0.08 (0.05, 0.11) |
| **Drug and alcohol episode** | | | | |

*(Continued)*

**Table 4.** (Continued)

| Variable | Odds Ratio (95% CI) | SE | p value | Marginal likelihood (95% CI) |
|---|---|---|---|---|
| No | *Reference* | . . . | . . . | 0.30 (0.29, 0.30) |
| Yes | 0.68 (0.60, 0.76) | 0.04 | <0.001 | 0.25 (0.24, 0.26) |
| **Presence of Autism Spectrum Disorder** | | | | |
| No | *Reference* | . . . | . . . | 0.29 (0.28, 0.30) |
| Yes | 1.62 (1.44, 1.83) | 0.10 | <0.001 | 0.35 (0.34, 0.37) |
| **Urgent admission** | | | | |
| No | *Reference* | . . . | . . . | 0.27 (0.27, 0.28) |
| Yes | 1.37 (1.28, 1.46) | 0.05 | <0.001 | 0.31 (0.31, 0.32) |
| **Summed Elix-Hauser comorbidities** | 1.49 (1.45, 1.54) | 0.02 | <0.001 | . . . |
| 0 | . . . | . . . | . . . | 0.26 (0.25, 0.26) |
| 1 | . . . | . . . | . . . | 0.31 (0.30, 0.31) |
| 2 | . . . | . . . | . . . | 0.36 (0.35, 0.37) |
| 3 | . . . | . . . | . . . | 0.42 (0.41, 0.43) |
| **Hospital type** | | | | |
| Public | *Reference* | . . . | . . . | 0.31 (0.30, 0.31) |
| Private | 0.26 (0.23, 0.30) | 0.02 | <0.001 | 0.16 (0.15, 0.17) |
| **Diagnosis chapter** | . . . | . . . | <0.001 | . . . |
| 5. Mental and behavioural | *Reference* | . . . | . . . | 0.41 (0.40, 0.43) |
| 1. Infectious and parasitic | 0.33 (0.28, 0.39) | 0.03 | <0.001 | 0.26 (0.24, 0.28) |
| 2. Neoplasms | 0.22 (0.18, 0.26) | 0.02 | <0.001 | 0.21 (0.20, 0.23) |
| 3. Blood and blood forming | 0.19 (0.14, 0.25) | 0.03 | <0.001 | 0.20 (0.17, 0.23) |
| 4. Endocrine | 0.37 (0.31, 0.44) | 0.03 | <0.001 | 0.28 (0.25, 0.30) |
| 6. Nervous | 0.33 (0.29, 0.37) | 0.02 | <0.001 | 0.26 (0.25, 0.27) |
| 7. Eye and adnexa | 0.31 (0.25, 0.37) | 0.03 | <0.001 | 0.25 (0.23, 0.28) |
| 8. Ear and mastoid | 0.34 (0.24, 0.48) | 0.06 | <0.001 | 0.27 (0.22, 0.31) |
| 9. Circulatory | 0.28 (0.24, 0.34) | 0.02 | <0.001 | 0.24 (0.23, 0.26) |
| 10. Respiratory | 0.46 (0.41, 0.52) | 0.03 | <0.001 | 0.30 (0.29, 0.32) |
| 11. Digestive | 0.44 (0.39, 0.48) | 0.02 | <0.001 | 0.30 (0.29, 0.31) |
| 12. Skin and subcutaneous | 0.27 (0.23, 0.32) | 0.02 | <0.001 | 0.24 (0.22, 0.26) |
| 13. Musculoskeletal | 0.31 (0.26, 0.36) | 0.03 | <0.001 | 0.25 (0.23, 0.27) |
| 14. Genitourinary | 0.31 (0.27, 0.36) | 0.02 | <0.001 | 0.25 (0.24, 0.27) |
| 15. Pregnancy and the puerperium | 0.34 (0.25, 0.44) | 0.05 | <0.001 | 0.26 (0.23, 0.30) |
| 17. Congenital and chromosomal | 1.11 (0.81, 1.52) | 0.18 | 0.504 | 0.43 (0.38, 0.47) |
| 18. Abnormal signs and symptoms | 0.33 (0.30, 0.37) | 0.02 | <0.001 | 0.26 (0.25, 0.27) |
| 19. Injury and poisoning | 0.46 (0.42, 0.51) | 0.02 | <0.001 | 0.31 (0.29, 0.32) |
| 21. Factors influencing contact | 0.43 (0.38, 0.48) | 0.03 | <0.001 | 0.29 (0.28, 0.31) |

SE: Standard error.

Overall, recognition of intellectual disability was low at only 23.79% of all episodes. Further, recognition of intellectual disability decreased over time, as evidenced by both the raw number of episodes where intellectual disability was recognised, and from the multi-level logistic regression controlling for other demographic and health variables. In contrast, recognition of intellectual disability increased the longer a person had been known to disability services. That is, though individuals were less likely overall to be recognised as having intellectual disability in the later years of the study period, those that had been known to disability services for a longer time were more likely to be recognised than those that had been known for a short time in

**Table 5. Subset analysis of predictors of intellectual disability recognition in a hospital episode for people with index dates after they turn 18 (n = 10,473).**

| Variable | Incident Rate Ratio (95% CI) | SE | p value | Marginal length of stay (95% CI) |
|---|---|---|---|---|
| **Sex** | | | | |
| Male | *Reference* | . . . | . . . | 0.30 (0.30, 0.31) |
| Female | 1.05 (0.96, 1.16) | 0.05 | 0.277* | 0.31 (0.30, 0.32) |
| **Age (years)** | 1.01 (1.01, 1.01) | < .01 | <0.001 | . . . |
| 20 | . . . | . . . | . . . | 0.29 (0.27, 0.30) |
| 30 | . . . | . . . | . . . | 0.29 (0.29, 0.30) |
| 40 | . . . | . . . | . . . | 0.30 (0.30, 0.31) |
| 50 | . . . | . . . | . . . | 0.31 (0.31, 0.32) |
| 60 | . . . | . . . | . . . | 0.32 (0.31, 0.33) |
| 70 | . . . | . . . | . . . | 0.33 (0.32, 0.35) |
| 80 | . . . | . . . | . . . | 0.34 (0.32, 0.36) |
| **Financial year** | . . . | . . . | <0.001 | . . . |
| 2005–2006 | *Reference* | . . . | . . . | 0.56 (0.54, 0.58) |
| 2006–2007 | 0.90 (0.79, 1.02) | 0.06 | 0.096 | 0.55 (0.53, 0.56) |
| 2007–2008 | 0.61 (0.53, 0.70) | 0.04 | <0.001 | 0.49 (0.47, 0.50) |
| 2008–2009 | 0.35 (0.30, 0.40) | 0.02 | <0.001 | 0.40 (0.39, 0.42) |
| 2009–2010 | 0.25 (0.22, 0.29) | 0.02 | <0.001 | 0.36 (0.34, 0.37) |
| 2010–2011 | 0.16 (0.14, 0.19) | 0.01 | <0.001 | 0.30 (0.28, 0.31) |
| 2011–2012 | 0.11 (0.09, 0.13) | 0.01 | <0.001 | 0.25 (0.24, 0.26) |
| 2012–2013 | 0.09 (0.08, 0.11) | 0.01 | <0.001 | 0.23 (0.22, 0.24) |
| 2013–2014 | 0.07 (0.06, 0.09) | 0.01 | <0.001 | 0.20 (0.19, 0.21) |
| 2014–2015 | 0.05 (0.04, 0.06) | 0.01 | <0.001 | 0.16 (0.15, 0.17) |
| **Years after recognition** | 1.27 (1.24, 1.30) | 0.01 | <0.001 | . . . |
| 0 | . . . | . . . | . . . | 0.20 (0.19, 0.21) |
| 2 | . . . | . . . | . . . | 0.25 (0.24, 0.26) |
| 4 | . . . | . . . | . . . | 0.31 (0.30, 0.31) |
| 6 | . . . | . . . | . . . | 0.37 (0.36, 0.38) |
| 8 | . . . | . . . | . . . | 0.43 (0.42, 0.45) |
| 10 | . . . | . . . | . . . | 0.50 (0.48, 0.52) |
| **Local Health District location** | . . . | . . . | <0.001 | . . . |
| Metropolitan | *Reference* | . . . | . . . | 0.29 (0.28, 0.30) |
| Rural | 1.59 (1.42, 1.77) | 0.09 | <0.001 | 0.35 (0.34, 0.36) |
| Speciality | 0.38 (0.30, 0.49) | 0.05 | <0.001 | 0.18 (0.15, 0.20) |
| **Remoteness** | . . . | . . . | <0.001 | . . . |
| Major cities | *Reference* | . . . | . . . | 0.33 (0.32, 0.33) |
| Inner regional | 0.66 (0.59, 0.74) | 0.04 | <0.001 | 0.27 (0.26, 0.28) |
| Outer regional and beyond | 0.46 (0.38, 0.55) | 0.04 | <0.001 | 0.23 (0.21, 0.25) |
| **IRSD quintile** | . . . | . . . | 0.110 | . . . |
| 1 (Most disadvantaged) | *Reference* | . . . | . . . | 0.30 (0.29, 0.32) |
| 2 | 1.04 (0.94, 1.15) | 0.05 | 0.451 | 0.31 (0.30, 0.32) |
| 3 | 0.98 (0.88, 1.1) | 0.05 | 0.771 | 0.30 (0.29, 0.31) |
| 4 | 1.14 (1.00, 1.29) | 0.07 | 0.046 | 0.32 (0.31, 0.34) |
| 5 (Least disadvantaged) | 0.97 (0.85, 1.11) | 0.07 | 0.678 | 0.30 (0.29, 0.31) |
| **Custody flag** | | | | |
| No | *Reference* | . . . | . . . | 0.31 (0.3, 0.32) |
| Yes | 0.09 (0.05, 0.17) | 0.03 | <0.001 | 0.08 (0.05, 0.12) |
| **Drug and alcohol flag** | | | | |

*(Continued)*

**Table 5.** (Continued)

| Variable | Incident Rate Ratio (95% CI) | SE | p value | Marginal length of stay (95% CI) |
|---|---|---|---|---|
| No | *Reference* | . . . | . . . | 0.31 (0.30, 0.32) |
| Yes | 0.61 (0.54, 0.70) | 0.04 | <0.001 | 0.25 (0.23, 0.26) |
| **Autism Spectrum Disorder flag** | | | | |
| No | *Reference* | . . . | . . . | 0.30 (0.29, 0.31) |
| Yes | 1.55 (1.35, 1.78) | 0.11 | <0.001 | 0.36 (0.34, 0.38) |
| **Emergency admission flag** | | | | |
| No | *Reference* | . . . | . . . | 0.28 (0.28, 0.29) |
| Yes | 1.5 (1.45, 1.55) | 0.02 | <0.001 | 0.33 (0.32, 0.34) |
| **Summed Elix-Hauser comorbidities** | 1.39 (1.30, 1.49) | 0.05 | <0.001 | . . . |
| 0 | . . . | . . . | . . . | 0.26 (0.26, 0.27) |
| 1 | . . . | . . . | . . . | 0.32 (0.31, 0.32) |
| 2 | . . . | . . . | . . . | 0.37 (0.36, 0.38) |
| 3 | . . . | . . . | . . . | 0.43 (0.42, 0.45) |
| **Hospital type** | | | | |
| Public | *Reference* | . . . | . . . | 0.32 (0.31, 0.32) |
| Private | 0.27 (0.24, 0.31) | 0.02 | <0.001 | 0.17 (0.16, 0.18) |
| **Diagnosis chapter** | . . . | . . . | <0.001 | . . . |
| 5. Mental and behavioural | *Reference* | . . . | . . . | 0.42 (0.41, 0.43) |
| 1. Infectious and parasitic | 0.36 (0.30, 0.43) | 0.03 | <0.001 | 0.28 (0.26, 0.30) |
| 2. Neoplasms | 0.22 (0.18, 0.27) | 0.02 | <0.001 | 0.22 (0.2, 0.24) |
| 3. Blood and blood forming | 0.19 (0.14, 0.25) | 0.03 | <0.001 | 0.20 (0.17, 0.23) |
| 4. Endocrine | 0.39 (0.32, 0.47) | 0.04 | <0.001 | 0.29 (0.27, 0.31) |
| 6. Nervous | 0.33 (0.29, 0.38) | 0.02 | <0.001 | 0.27 (0.26, 0.28) |
| 7. Eye and adnexa | 0.31 (0.26, 0.39) | 0.03 | <0.001 | 0.26 (0.24, 0.29) |
| 8. Ear and mastoid | 0.32 (0.22, 0.46) | 0.06 | <0.001 | 0.26 (0.22, 0.31) |
| 9. Circulatory | 0.29 (0.24, 0.34) | 0.03 | <0.001 | 0.25 (0.23, 0.27) |
| 10. Respiratory | 0.48 (0.42, 0.55) | 0.03 | <0.001 | 0.32 (0.30, 0.33) |
| 11. Digestive | 0.45 (0.40, 0.50) | 0.03 | <0.001 | 0.31 (0.30, 0.32) |
| 12. Skin and subcutaneous | 0.30 (0.26, 0.36) | 0.03 | <0.001 | 0.26 (0.24, 0.28) |
| 13. Musculoskeletal | 0.31 (0.26, 0.38) | 0.03 | <0.001 | 0.26 (0.24, 0.28) |
| 14. Genitourinary | 0.33 (0.29, 0.39) | 0.03 | <0.001 | 0.27 (0.25, 0.29) |
| 15. Pregnancy and the puerperium | 0.38 (0.27, 0.52) | 0.06 | <0.001 | 0.29 (0.24, 0.33) |
| 17. Congenital and chromosomal | 1.23 (0.86, 1.77) | 0.23 | 0.264 | 0.45 (0.40, 0.51) |
| 18. Abnormal signs and symptoms | 0.33 (0.30, 0.37) | 0.02 | <0.001 | 0.27 (0.26, 0.28) |
| 19. Injury and poisoning | 0.50 (0.44, 0.55) | 0.03 | <0.001 | 0.32 (0.31, 0.33) |
| 21. Factors influencing contact | 0.45 (0.40, 0.52) | 0.03 | <0.001 | 0.31 (0.29, 0.32) |

SE: Standard error.

*Indicates difference in significance from main analysis

those later years. These patterns of recognition in an acute health care setting over time and as a function of recency of intellectual disability diagnosis in disability services are concerning, particularly as they appear to be at loggerheads with attempts to improve integration between health and disability services in NSW in that period [35, 36].

Despite low overall recognition of intellectual disability, one encouraging aspect of our results is that they suggest adults with more complex health needs are more likely to be recognised as having intellectual disability during an inpatient stay. Specifically, Autistic

**Table 6. Predictors of length of hospital episode in days (n = 12,593).**

| Variable | Incident Rate Ratio (95% CI) | SE | p value | Marginal length of stay (95% CI) |
|---|---|---|---|---|
| **Intellectual disability recognition** | | | | |
| No | *Reference* | | . . . | 3.94 (3.81, 4.07) |
| Yes | 1.57 (1.56, 1.59) | 0.01 | <0.001 | 6.20 (6.00, 6.41) |
| **Sex** | | | | |
| Male | *Reference* | . . . | . . . | 5.15 (4.96, 5.34) |
| Female | 0.68 (0.66, 0.70) | 0.01 | <0.001 | 3.51 (3.39, 3.64) |
| **Age (years)** | 0.99 (0.99, 0.99) | < .01 | <0.001 | . . . |
| 20 | . . . | . . . | . . . | 5.42 (5.18, 5.65) |
| 30 | . . . | . . . | . . . | 4.86 (4.69, 5.04) |
| 40 | . . . | . . . | . . . | 4.37 (4.23, 4.51) |
| 50 | . . . | . . . | . . . | 3.92 (3.8, 4.05) |
| 60 | . . . | . . . | . . . | 3.53 (3.40, 3.65) |
| 70 | . . . | . . . | . . . | 3.17 (3.02, 3.31) |
| 80 | . . . | . . . | . . . | 2.84 (2.69, 3.00) |
| **Financial year** | . . . | . . . | <0.001 | . . . |
| 2005–2006 | *Reference* | . . . | . . . | 4.34 (4.1, 4.59) |
| 2006–2007 | 1.00 (0.98, 1.03) | 0.01 | 0.742 | 4.36 (4.14, 4.59) |
| 2007–2008 | 1.04 (1.01, 1.07) | 0.01 | 0.007 | 4.51 (4.30, 4.72) |
| 2008–2009 | 1.06 (1.02, 1.09) | 0.02 | 0.001 | 4.59 (4.40, 4.78) |
| 2009–2010 | 1.12 (1.08, 1.16) | 0.02 | <0.001 | 4.85 (4.67, 5.03) |
| 2010–2011 | 1.20 (1.15, 1.25) | 0.03 | <0.001 | 5.20 (5.02, 5.38) |
| 2011–2012 | 1.04 (0.99, 1.09) | 0.03 | 0.161 | 4.50 (4.34, 4.65) |
| 2012–2013 | 0.97 (0.92, 1.02) | 0.03 | 0.280 | 4.22 (4.07, 4.36) |
| 2013–2014 | 0.92 (0.86, 0.98) | 0.03 | 0.006 | 3.99 (3.85, 4.14) |
| 2014–2015 | 0.97 (0.91, 1.04) | 0.03 | 0.369 | 4.21 (4.04, 4.38) |
| **Years after recognition** | 1.06 (1.05, 1.07) | < .01 | <0.001 | . . . |
| 0 | . . . | . . . | . . . | 3.47 (3.33, 3.61) |
| 2 | . . . | . . . | . . . | 3.90 (3.77, 4.04) |
| 4 | . . . | . . . | . . . | 4.39 (4.24, 4.53) |
| 6 | . . . | . . . | . . . | 4.93 (4.75, 5.12) |
| 8 | . . . | . . . | . . . | 5.55 (5.29, 5.81) |
| 10 | . . . | . . . | . . . | 6.24 (5.88, 6.60) |
| **Local Health District location** | . . . | . . . | <0.001 | . . . |
| Metropolitan | *Reference* | . . . | . . . | 4.18 (4.04, 4.32) |
| Rural | 1.15 (1.12, 1.17) | 0.01 | <0.001 | 4.79 (4.62, 4.96) |
| Specialty | 1.20 (1.17, 1.23) | 0.02 | <0.001 | 5.03 (4.83, 5.23) |
| **Remoteness** | . . . | . . . | <0.001 | . . . |
| Major cities | *Reference* | . . . | . . . | 4.37 (4.22, 4.52) |
| Inner regional | 1.07 (1.05, 1.10) | 0.01 | <0.001 | 4.69 (4.52, 4.87) |
| Outer regional and beyond | 0.96 (0.93, 1.01) | 0.02 | 0.089 | 4.22 (4.01, 4.42) |
| **IRSD quintile** | . . . | . . . | <0.001 | . . . |
| 1 (Most disadvantaged) | *Reference* | . . . | . . . | 3.90 (3.77, 4.03) |
| 2 | 1.11 (1.08, 1.13) | 0.01 | <0.001 | 4.31 (4.16, 4.46) |
| 3 | 1.32 (1.29, 1.34) | 0.01 | <0.001 | 5.13 (4.95, 5.31) |
| 4 | 1.41 (1.38, 1.44) | 0.02 | <0.001 | 5.49 (5.29, 5.70) |
| 5 (Least disadvantaged) | 1.07 (1.04, 1.09) | 0.01 | <0.001 | 4.16 (4.01, 4.31) |
| **In custody at time of episode** | | | | |

*(Continued)*

**Table 6.** (Continued)

| Variable | Incident Rate Ratio (95% CI) | SE | p value | Marginal length of stay (95% CI) |
|---|---|---|---|---|
| No | *Reference* | . . . | . . . | 4.25 (4.11, 4.38) |
| Yes | 2.95 (2.85, 3.06) | 0.05 | <0.001 | 12.55 (11.97, 13.13) |
| **Drug and alcohol episode** | | | | |
| No | *Reference* | . . . | . . . | 4.43 (4.28, 4.57) |
| Yes | 1 (0.99, 1.02) | 0.01 | 0.547 | 4.45 (4.29, 4.61) |
| **Presence of Autism Spectrum Disorder** | | | | |
| No | *Reference* | . . . | . . . | 4.14 (4.01, 4.28) |
| Yes | 1.77 (1.66, 1.89) | 0.06 | <0.001 | 7.33 (6.82, 7.85) |
| **Urgent admission** | | | | |
| No | *Reference* | . . . | . . . | 4.82 (4.66, 4.98) |
| Yes | 0.85 (0.84, 0.86) | < .01 | <0.001 | 4.10 (3.96, 4.23) |
| **Summed Elix-Hauser comorbidities** | 1.56 (1.55, 1.57) | < .01 | <0.001 | . . . |
| 0 | . . . | . . . | . . . | 2.88 (2.78, 2.97) |
| 1 | . . . | . . . | . . . | 4.49 (4.34, 4.63) |
| 2 | . . . | . . . | . . . | 7.00 (6.77, 7.23) |
| 3 | . . . | . . . | . . . | 10.92 (10.55, 11.29) |
| **Hospital type** | | | | |
| Public | *Reference* | . . . | . . . | 4.48 (4.33, 4.63) |
| Private | 0.80 (0.79, 0.82) | 0.01 | <0.001 | 3.60 (3.46, 3.74) |
| **Diagnosis chapter** | . . . | . . . | <0.001 | . . . |
| 5. Mental and behavioural | *Reference* | . . . | . . . | 6.49 (6.28, 6.71) |
| 1. Infectious and parasitic | 0.51 (0.49, 0.52) | 0.01 | <0.001 | 3.30 (3.16, 3.43) |
| 2. Neoplasms | 0.45 (0.43, 0.46) | 0.01 | <0.001 | 2.89 (2.77, 3.01) |
| 3. Blood and blood forming | 0.35 (0.33, 0.36) | 0.01 | <0.001 | 2.25 (2.12, 2.38) |
| 4. Endocrine | 0.48 (0.47, 0.50) | 0.01 | <0.001 | 3.12 (2.99, 3.25) |
| 6. Nervous | 0.39 (0.38, 0.40) | < .01 | <0.001 | 2.55 (2.46, 2.64) |
| 7. Eye and adnexa | 0.29 (0.28, 0.31) | 0.01 | <0.001 | 1.89 (1.78, 1.99) |
| 8. Ear and mastoid | 0.32 (0.29, 0.35) | 0.02 | <0.001 | 2.07 (1.85, 2.30) |
| 9. Circulatory | 0.43 (0.42, 0.45) | 0.01 | <0.001 | 2.81 (2.70, 2.93) |
| 10. Respiratory | 0.57 (0.56, 0.58) | 0.01 | <0.001 | 3.68 (3.56, 3.81) |
| 11. Digestive | 0.39 (0.38, 0.39) | < .01 | <0.001 | 2.51 (2.42, 2.6) |
| 12. Skin and subcutaneous | 0.68 (0.66, 0.70) | 0.01 | <0.001 | 4.42 (4.24, 4.59) |
| 13. Musculoskeletal | 0.56 (0.55, 0.58) | 0.01 | <0.001 | 3.66 (3.51, 3.81) |
| 14. Genitourinary | 0.45 (0.44, 0.46) | 0.01 | <0.001 | 2.90 (2.79, 3.02) |
| 15. Pregnancy and the puerperium | 0.74 (0.70, 0.78) | 0.02 | <0.001 | 4.79 (4.51, 5.06) |
| 17. Congenital and chromosomal | 0.71 (0.67, 0.75) | 0.02 | <0.001 | 4.61 (4.32, 4.90) |
| 18. Abnormal signs and symptoms | 0.35 (0.34, 0.36) | < .01 | <0.001 | 2.28 (2.20, 2.36) |
| 19. Injury and poisoning | 0.62 (0.61, 0.63) | 0.01 | <0.001 | 4.05 (3.91, 4.19) |
| 21. Factors influencing contact | 1.06 (1.04, 1.08) | 0.01 | <0.001 | 6.89 (6.66, 7.13) |

SE: Standard Error.

people, people with high Elix-Hauser comorbidities score recorded in an episode, or people visiting hospital for an urgent admission were more likely to be recognised as having an intellectual disability. These factors may reflect that people with more obvious disability and complex needs are more likely to be recognised as having intellectual disability than people with milder intellectual disability and correspondingly lower support needs [20]. For

**Table 7. Subset analysis of predictors of length of hospital episode in days for people with index dates after they turn 18 (n = 10,473).**

| Variable | Incident Rate Ratio (95% CI) | SE | p value | Marginal length of stay (95% CI) |
|---|---|---|---|---|
| **Intellectual disability recognition** | | | | |
| No | *Reference* | | . . . | 4.42 (4.23, 4.61) |
| Yes | 1.59 (1.57, 1.60) | 0.01 | <0.001 | 7.02 (6.72, 7.31) |
| **Sex** | | | | |
| Male | *Reference* | . . . | . . . | 5.88 (5.61, 6.14) |
| Female | 0.65 (0.63, 0.67) | 0.01 | <0.001 | 3.81 (3.65, 3.98) |
| **Age (years)** | 0.99 (0.99, 0.99) | < .01 | <0.001 | |
| 20 | . . . | . . . | . . . | 6.22 (5.86, 6.58) |
| 30 | . . . | . . . | . . . | 5.59 (5.32, 5.85) |
| 40 | . . . | . . . | . . . | 5.02 (4.81, 5.22) |
| 50 | . . . | . . . | . . . | 4.51 (4.33, 4.68) |
| 60 | . . . | . . . | . . . | 4.05 (3.88, 4.22) |
| 70 | . . . | . . . | . . . | 3.64 (3.46, 3.82) |
| 80 | . . . | . . . | . . . | 3.27 (3.07, 3.46) |
| **Financial year** | . . . | . . . | <0.001 | |
| 2005–2006 | *Reference* | . . . | . . . | 4.92 (4.6, 5.25) |
| 2006–2007 | 1.01 (0.98, 1.04) | 0.01 | 0.429 | 4.98 (4.68, 5.27) |
| 2007–2008 | 1.05 (1.02, 1.08) | 0.02 | 0.000 | 5.19 (4.91, 5.47) |
| 2008–2009 | 1.04 (1.01, 1.08) | 0.02 | 0.020 | 5.12 (4.87, 5.38) |
| 2009–2010 | 1.12 (1.07, 1.16) | 0.02 | <0.001 | 5.49 (5.24, 5.75) |
| 2010–2011 | 1.21 (1.16, 1.26) | 0.03 | <0.001 | 5.95 (5.69, 6.21) |
| 2011–2012 | 1.02 (0.97, 1.08) | 0.03 | 0.397 | 5.04 (4.82, 5.25) |
| 2012–2013 | 0.97 (0.91, 1.02) | 0.03 | 0.234 | 4.75 (4.55, 4.96) |
| 2013–2014 | 0.89 (0.84, 0.95) | 0.03 | 0.001 | 4.40 (4.20, 4.60) |
| 2014–2015 | 0.92 (0.86, 0.99) | 0.03 | 0.029 | 4.54 (4.32, 4.76) |
| **Years after recognition** | 1.06 (1.05, 1.07) | < .01 | <0.001 | |
| 0 | . . . | . . . | . . . | 3.90 (3.71, 4.09) |
| 2 | . . . | . . . | . . . | 4.41 (4.22, 4.60) |
| 4 | . . . | . . . | . . . | 4.99 (4.78, 5.20) |
| 6 | . . . | . . . | . . . | 5.64 (5.37, 5.91) |
| 8 | . . . | . . . | . . . | 6.37 (6.01, 6.74) |
| 10 | . . . | . . . | . . . | 7.21 (6.71, 7.71) |
| **Local Health District location** | . . . | . . . | <0.001 | |
| Metropolitan | *Reference* | . . . | . . . | 4.64 (4.45, 4.84) |
| Rural | 1.19 (1.17, 1.22) | 0.01 | <0.001 | 5.55 (5.30, 5.79) |
| Speciality | 1.11 (1.08, 1.14) | 0.02 | <0.001 | 5.16 (4.91, 5.42) |
| **Remoteness** | . . . | . . . | <0.001 | |
| Major cities | *Reference* | . . . | . . . | 4.88 (4.67, 5.09) |
| Inner regional | 1.10 (1.07, 1.13) | 0.02 | <0.001 | 5.35 (5.10, 5.59) |
| Outer regional and beyond | 0.95 (0.91, 1.00) | 0.02 | 0.029* | 4.65 (4.39, 4.91) |
| **IRSD quintile** | . . . | . . . | <0.001 | |
| 1 (Most disadvantaged) | *Reference* | . . . | . . . | 4.25 (4.07, 4.43) |
| 2 | 1.11 (1.09, 1.13) | 0.01 | <0.001 | 4.72 (4.51, 4.93) |
| 3 | 1.40 (1.38, 1.43) | 0.01 | <0.001 | 5.97 (5.7, 6.23) |
| 4 | 1.51 (1.47, 1.55) | 0.02 | <0.001 | 6.42 (6.12, 6.71) |
| 5 (Least disadvantaged) | 1.11 (1.08, 1.14) | 0.01 | <0.001 | 4.72 (4.51, 4.93) |
| **Custody flag** | | | | |

*(Continued)*

**Table 7.** (*Continued*)

| Variable | Incident Rate Ratio (95% CI) | SE | p value | Marginal length of stay (95% CI) |
|---|---|---|---|---|
| No | *Reference* | . . . | . . . | 4.74 (4.55, 4.93) |
| Yes | 3.1 (2.99, 3.21) | 0.06 | <0.001 | 14.69 (13.91, 15.47) |
| **Drug and alcohol flag** | | | | |
| No | *Reference* | . . . | . . . | 4.97 (4.76, 5.18) |
| Yes | 1.01 (0.99, 1.02) | 0.01 | 0.533 | 5.00 (4.77, 5.22) |
| **Autism Spectrum Disorder flag** | | | | |
| No | *Reference* | . . . | . . . | 4.61 (4.42, 4.8) |
| Yes | 2.08 (1.91, 2.26) | 0.09 | <0.001 | 9.58 (8.72, 10.45) |
| **Emergency admission flag** | | | | |
| No | *Reference* | . . . | . . . | 5.42 (5.19, 5.65) |
| Yes | 0.84 (0.83, 0.85) | < .01 | <0.001 | 4.56 (4.37, 4.76) |
| **Summed Elix-Hauser comorbidities** | 1.56 (1.55, 1.56) | < .01 | <0.001 | |
| 0 | . . . | . . . | . . . | 3.18 (3.05, 3.31) |
| 1 | . . . | . . . | . . . | 4.96 (4.75, 5.16) |
| 2 | . . . | . . . | . . . | 7.72 (7.39, 8.04) |
| 3 | . . . | . . . | . . . | 12.03 (11.51, 12.54) |
| **Hospital type** | | | | |
| Public | *Reference* | . . . | . . . | 5.03 (4.82, 5.24) |
| Private | 0.77 (0.76, 0.79) | 0.01 | 0.000 | 3.90 (3.71, 4.08) |
| **Diagnosis chapter** | . . . | . . . | <0.001 | |
| 5. Mental and behavioural | *Reference* | . . . | . . . | 7.33 (7.02, 7.64) |
| 1. Infectious and parasitic | 0.49 (0.48, 0.51) | 0.01 | <0.001 | 3.60 (3.42, 3.78) |
| 2. Neoplasms | 0.42 (0.41, 0.43) | 0.01 | <0.001 | 3.08 (2.93, 3.23) |
| 3. Blood and blood forming | 0.33 (0.31, 0.35) | 0.01 | <0.001 | 2.43 (2.28, 2.59) |
| 4. Endocrine | 0.47 (0.45, 0.48) | 0.01 | <0.001 | 3.43 (3.26, 3.60) |
| 6. Nervous | 0.37 (0.36, 0.38) | < .01 | <0.001 | 2.73 (2.61, 2.85) |
| 7. Eye and adnexa | 0.28 (0.27, 0.30) | 0.01 | <0.001 | 2.08 (1.94, 2.21) |
| 8. Ear and mastoid | 0.31 (0.27, 0.34) | 0.02 | <0.001 | 2.25 (1.98, 2.51) |
| 9. Circulatory | 0.42 (0.41, 0.43) | 0.01 | <0.001 | 3.09 (2.94, 3.23) |
| 10. Respiratory | 0.55 (0.54, 0.56) | 0.01 | <0.001 | 4.04 (3.86, 4.21) |
| 11. Digestive | 0.37 (0.37, 0.38) | < .01 | <0.001 | 2.74 (2.63, 2.86) |
| 12. Skin and subcutaneous | 0.66 (0.64, 0.68) | 0.01 | <0.001 | 4.82 (4.59, 5.05) |
| 13. Musculoskeletal | 0.56 (0.55, 0.58) | 0.01 | <0.001 | 4.12 (3.92, 4.32) |
| 14. Genitourinary | 0.44 (0.43, 0.45) | 0.01 | <0.001 | 3.22 (3.07, 3.37) |
| 15. Pregnancy and the puerperium | 0.68 (0.64, 0.72) | 0.02 | <0.001 | 4.97 (4.63, 5.32) |
| 17. Congenital and chromosomal | 0.60 (0.56, 0.64) | 0.02 | <0.001 | 4.38 (4.04, 4.72) |
| 18. Abnormal signs and symptoms | 0.34 (0.33, 0.35) | < .01 | <0.001 | 2.48 (2.38, 2.59) |
| 19. Injury and poisoning | 0.62 (0.61, 0.63) | 0.01 | <0.001 | 4.56 (4.37, 4.76) |
| 21. Factors influencing contact | 1.05 (1.03, 1.07) | 0.01 | <0.001 | 7.71 (7.38, 8.04) |

SE: Standard error.

*Indicates difference in significance from main analysis

example, Autism Spectrum Disorders are more common in those with a greater level of intellectual disability [27].

Though complex health needs predicted intellectual disability recognition, we found that episodes with complexity across other domains, such as episodes with drug and alcohol related

codes, or episodes that occurred while the individual was in custody, predicted a decreased chance of intellectual disability recognition. This difference was particularly stark for individuals in custody, where fewer than 10% of episodes that occurred when the individual was in custody recognised intellectual disability. Our findings are particularly concerning given the over-representation of individuals with intellectual disability in custody, and their need for extra support (not less) to prevent recidivism [21]. Overall, the results suggest a blindness to the presence of intellectual disability in such individuals, risking lack of activation of supports in areas that are critical to their future trajectory.

Our findings complement and extend those of Bourke et al., who examined predictors of intellectual disability recognition in children in Western Australia [20]. Similar to their findings, our studied showed that females admitted to hospital were more likely to be recognised as having intellectual disability than males, though we note that in our study the effect of sex was not significant when we excluded adults who first appeared in the DS-MDS, PG, or SDS prior to turning 18. Furthermore, both our study and their study indicated that those with severe intellectual disability were more likely to be recognised as having an intellectual disability in hospital (suggested by a greater likelihood for Autistic people, people with high Elix-Hauser comorbidities score recorded in an episode, or people visiting hospital for an urgent admission to be recognised as having intellectual disability). We extend the work of Bourke et al. through the addition of more variables about an individual's profile (such as whether the person is Autistic, or currently in custody), as well as the addition of variables surrounding the hospital itself during an admission (such as whether the hospital was in a rural or metropolitan area, and whether it was a public or private hospital). Overall, our work expands our understanding of the recognition of intellectual disability in hospitals to adults, and shows that there are notable similarities between children and adults when it comes to the recognition of intellectual disability in hospital.

For length of stay, episodes where intellectual disability was recognised were around 60% longer than episodes where it was not. It is likely that those with more complex health needs are more likely to both require longer inpatient stays and be recognised as having intellectual disability. Extended episode length could be mitigated by the activation of reasonable adjustments, which may assist more assertive addressing of complex health needs and coordinate supports required for earlier discharge. Our data points do not afford us knowledge of whether adjustments were applied and nor is their application mandatory within the health service system in Australia. A requirement to both identify those with intellectual disability, and implement reasonable adjustments would afford potential to optimise the health care journey for a person with intellectual disability. Should such data on mandatory adjustments be available to researchers, the impact of this strategy on the health care experience or people with intellectual disability could be evaluated from multiple perspectives.

Taken together, our results support previous research on the need for more consistent recognition of intellectual disability when receiving health services [11, 20]. Low recognition rates of intellectual disability highlight a major barrier to providing reasonable adjustments in health care settings for people with intellectual disability, particularly for those in contact with the justice system, and those presenting to hospital for reasons related to drug and alcohol use. To address the issue of low recognition, we advocate for the introduction of disability specific flags within hospital records that code for the presence, type, and severity of a person's disability when they present to hospital. Ideally, such a flag would be linked with national databases tracking such information for people with disability (such as the Reasonable Adjustments Flag in the United Kingdom, or the proposed National Disability Data Asset in Australia) [17, 37]. Overall, our results suggest that recognition of intellectual disability has been historically poor, and more needs to be done to assure that people with intellectual disability can receive the best possible health care when they come into contact with health services.

## Strengths and limitations

A key strength of this study was the ability to leverage a large amount of population-level hospital data from a broad period. However, this approach also has its limitations. Though we could identify from our dataset whether intellectual disability had been recorded as an ICD-10 code during an inpatient episode, the absence of this record does not necessarily guarantee that the hospital was not aware that an individual had an intellectual disability. However, we believe that it does provide a reasonable proxy for recognition of intellectual disability in the absence of other information, and feel this further highlights the need for a flag for intellectual disability to be included as standard in hospital records. Also, we were not able to directly determine the severity of an individual's intellectual disability, and as discussed it is likely that there is some overlap between the severity of the disability and the likelihood that it is recognised during an inpatient episode. However, this fact does not account for the low absolute amount of recognition across the episodes. Finally, the estimated marginal proportion of recognition per financial year in our model (Table 3) was different from the crude observed proportions despite showing the same trend (Fig 1). The difference is expected as the model controls for other variables that the crude proportion does not.

## Conclusions

Identification of intellectual disability during contact with health services is critical to providing reasonable adjustments to health provision to assure the best health outcomes for adults with intellectual disability. The current study examined demographic and health predictors of intellectual disability recognition in inpatient episodes in NSW, Australia. Overall, we find that complex health needs appear to predict the recognition of intellectual disability, those with complex needs across other domains (such as drug and alcohol problems, contacts with the justice system) have a suppressive effect on recognising intellectual disability, and recognition of intellectual disability was associated with longer hospital stays. Despite an increased likelihood of recognition for those with complex health care needs, overall rates of recognition (20% to 35%) are still unacceptably low. The introduction of targeted initiatives, such as the development of an intellectual disability specific flag in hospital records, may help improve the recognition of intellectual disability during contact with health services, and aid in the provision of reasonable adjustments for people with intellectual disability.

## Author Contributions

**Conceptualization:** Julian Norman Trollor, Tony Florio, Preeyaporn Srasuebkul.

**Data curation:** Preeyaporn Srasuebkul.

**Formal analysis:** Adrian Raymond Walker.

**Funding acquisition:** Julian Norman Trollor, Tony Florio, Preeyaporn Srasuebkul.

**Investigation:** Adrian Raymond Walker, Preeyaporn Srasuebkul.

**Methodology:** Adrian Raymond Walker, Julian Norman Trollor, Tony Florio.

**Project administration:** Adrian Raymond Walker, Preeyaporn Srasuebkul.

**Resources:** Julian Norman Trollor.

**Software:** Adrian Raymond Walker, Preeyaporn Srasuebkul.

**Supervision:** Julian Norman Trollor, Preeyaporn Srasuebkul.

**Validation:** Adrian Raymond Walker, Tony Florio, Preeyaporn Srasuebkul.

**Visualization:** Adrian Raymond Walker.

**Writing – original draft:** Adrian Raymond Walker.

**Writing – review & editing:** Julian Norman Trollor, Tony Florio, Preeyaporn Srasuebkul.

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
