## [Decision Letter · Decision Letter 0]

23 Nov 2021

PONE-D-21-32313Predictors and outcomes of recognition of intellectual disability for adults during hospital admissions: A retrospective data linkage study in NSW, AustraliaPLOS ONE

Dear Dr. Trollor,

Thank you for submitting your manuscript to PLOS ONE. After careful consideration, we feel that it has merit but does not fully meet PLOS ONE’s publication criteria as it currently stands. Therefore, we invite you to submit a revised version of the manuscript that addresses the points raised during the review process.

We look forward to receiving your revised manuscript.

Kind regards,

Rashidul Alam Mahumud, MPH, MSc, PhD

Academic Editor

PLOS ONE

Journal Requirements:

“JT is funded as the Chair of Intellectual Disability Mental Health by the Mental Health Branch of the NSW Ministry of Health.”

 “This study was funded by a National Health and Medical Research Council Australia (https://www.nhmrc.gov.au/) funded Partnerships for Better Health grant (APP1056128; Title: Improving the Mental Health Outcomes of People with an Intellectual Disability; Awarded to: JT, EE, RL, LD, KF, AJ, KD) and a National Health and Medical Research Council Australia funded Project grant (APP1123033; Title: Understanding health service system needs for people with intellectual disability; Awarded to: JT, CV, NL, RM, SR). The funders had no role in study design, data collection and analysis, decision to publish, or preparation of the manuscript.”

Additional Editor Comments:

Please revise the manuscript according to the reviewer's comments and advise. Please see my some concerns as follow:

1) What is your rationale for selecting variables in Table 2?

2) Table 2 may be developed for considering all variables from Table 1.

3) What is your protocol for adding variables in the regression model?

4) It would be better more productive to add supplementary tables in the manuscript as follow Table 3.1-Table 3.3 (Table 3).

5) The discussion section can be revised to incorporate the comparisons and contrast findings.

Reviewers' comments:

Reviewer's Responses to Questions

**Comments to the Author**

1. Is the manuscript technically sound, and do the data support the conclusions?

Reviewer #1: Yes

2. Has the statistical analysis been performed appropriately and rigorously? 

Reviewer #1: Yes

3. Have the authors made all data underlying the findings in their manuscript fully available?

Reviewer #1: Yes

4. Is the manuscript presented in an intelligible fashion and written in standard English?

Reviewer #1: Yes

5. Review Comments to the Author

Reviewer #1: Thank you for allowing me to review this paper. This review paper looks unique and give us some insight into the issue. However, I have some observations regarding the manuscripts. My observations are;

Abstract:

• It would be great if the author could elaborate on the result of this study a bit.

Introduction:

Method:

• Definition of the study population is missing.

• The authors have not mentioned the type of this study.

• There are no details about the comparison group.

• What is the outcome variable?

• Was each case followed up till the outcome?

• Details on statistical analysis are not sufficient. If we followed up the participants till the outcome, should not be the Cox regression more appropriate?

Result:

• “intellectual disability was recognized in 19,261 (23.79%) of all episodes”- from what point does the author deduce that?

Discussion:

• The author thoroughly detailed the result section here. The author gave no perspective or comparison.

• The authors also failed to cite many related articles

6. PLOS authors have the option to publish the peer review history of their article (what does this mean?). If published, this will include your full peer review and any attached files.

Reviewer #1: No

---

## [Author Response · Author response to Decision Letter 0]

6 Jan 2022

Editor: 

1) What is your rationale for selecting variables in Table 2?

Response: Our variables were selected based on discussion between the authors (using expert knowledge and referring to previous literature) on what variables our dataset would be important in predicting the recognition of intellectual disability.

To clarify this in our manuscript, we have added a section explaining our rationale for variable selection in the Methods section on pages 9 & 10. We have also included references to additional literature that were used when selecting our variables. The section now reads:

Fixed effects for the multi-level regression can be seen in Table 1. From the variables available in our datasets, we selected fixed effects for our models through a combination of expert medical knowledge about intellectual disability (JT), and internal discussion between the authors based on previous literature. [9,20,27] We included all variables selected in our analysis of intellectual disability recognition as fixed effects (besides intellectual disability recognition, which was the outcome variable for this model). Most fixed effects included were at the episode level (i.e., from data obtained directly from the records in that episode), with some variables (including age, Aboriginal and Torres Strait Islander Status, presence of Autism Spectrum Disorder) determined for each person based on the linkage method described in Reppermund et al.[22] We used ICD-10 chapter 5 as the reference level for the principal diagnosis chapter variable as most ICD-10 codes for intellectual disability are in this chapter (and may include cases where the principal diagnosis was intellectual disability). We included a random effect of person identifier into the model. We did not include any ICD10 chapters where there were no records.

2) Table 2 may be developed for considering all variables from Table 1.

Response: Currently, Table 2 includes only person level variables from Table 1 to give a snap-shot of the people in our cohort. 

We have now added a new Table 3 that details the episode level variables for all admissions in our cohort on page 16. 

3) What is your protocol for adding variables in the regression model?

Response: All variables that were selected as potentially relevant to our question based on internal discussion and prior literature were included in our regression model. As such, we hope this point is adequately addressed within our manuscript the changes in response to point 1. The variables in the regression model are select priori and we did not employ any statistical method for the variable selection.

4) It would be better more productive to add supplementary tables in the manuscript as follow Table 3.1-Table 3.3 (Table 3).

Response: We have added our tables previously in the supplementary to the manuscript (noting that we no longer have a supplementary document). We have included the full regression results from the length of stay analysis as its own table (rather than as Table 4.2) as it we felt that it was more natural in the flow of the manuscript. These new tables are on pages 22-23 & 25-29.

We have also included some additional exposition to better explain the tables on page 12. It reads:

To ensure that our results were not biased by including people who were identified in the DS-MDS, PG, or SDS as having intellectual disability before they turned 18, we conducted two subset analyses (one for intellectual disability recognition, and one for length of stay) restricting the study population to only individuals who were over 18 at their index date (i.e., when they first appeared in the DS-MDS, PG, or SDS). The results of these regressions are reported alongside our other analyses for comparison.

We have also mentioned the results of these regressions explicitly on pages 19 and 24.

5) The discussion section can be revised to incorporate the comparisons and contrast findings.

Response: We have expanded on our discussion on page 32 to compare and contrast our findings to those of Bourke et al., whose work we cite as a motivating paper in our introduction. The new paragraph in the discussion reads as below:

Our findings complement and extend those of Bourke et al., who examined predictors of intellectual disability recognition in children in Western Australia.[20] Similar to their findings, our studied showed that females admitted to hospital were more likely to be recognised as having intellectual disability than males, though we note that in our study the effect of sex was not significant when we excluded adults who first appeared in the DS-MDS, PG, or SDS prior to turning 18. Furthermore, both our study and their study indicated that those with severe intellectual disability were more likely to be recognised as having an intellectual disability in hospital (suggested by a greater likelihood for Autistic people, people with high Elix-Hauser comorbidities score recorded in an episode, or people visiting hospital for an urgent admission to be recognised as having intellectual disability). We extend the work of Bourke et al. through the addition of more variables about an individual’s profile (such as whether the person is Autistic, or currently in custody), as well as the addition of variables surrounding the hospital itself during an admission (such as whether the hospital was in a rural or metropolitan area, and whether it was a public or private hospital). Overall, our work expands our understanding of the recognition of intellectual disability in hospitals to adults, and shows that there are notable similarities between children and adults when it comes to the recognition of intellectual disability in hospital. 

Reviewer 1:

1) Abstract: It would be great if the author could elaborate on the result of this study a bit.

Response: We have attempted to add more to our abstract on our results, noting that we are already close to the abstract word limit. Our new abstract dedicates over one-third of its words to the results (starting from “We found an overall low rate of recognition…” to “Recognition of intellectual disability was associated with longer episodes of care…”, though this could be increased further at the discretion of the editor. The new abstract in full reads:

Adults with intellectual disability have high health care needs. Despite frequent contact with health services, they often receive inadequate health care. One method to improve health care delivery is reasonable adjustments, that is, the adaptation of health care delivery such that barriers to participation are removed for the person with disability. A starting point for the provision of reasonable adjustments is recognition of intellectual disability during the health care contact. To determine rates and predictors of the recognition of intellectual disability during hospital admissions, and its impact on admission metrics, we examined a population of adults with intellectual disability identified from disability services datasets from New South Wales, Australia between 2005 and 2014. Recognition of intellectual disability was determined by the recording of an International Classification of Diseases 10th revision (ICD-10) diagnostic code for intellectual disability during a given hospital admission. We examined how recognition of intellectual disability related to length of hospital episodes. We found an overall low rate of recognition of intellectual disability (23.79%) across all hospital episodes, with the proportion of hospital episodes recognising intellectual disability decreasing from 2005-2015. Admissions for adults with complex health profiles (e.g., those with many comorbidities, those with Autism Spectrum Disorder, and those admitted for urgent treatment) were more likely to recognise intellectual disability, but admissions for adults with complexity in other domains (i.e., for those in custody, or those with drug and alcohol disorders) were less likely to recognise intellectual disability. Recognition of intellectual disability was associated with longer episodes of care, possibly indicating the greater provision of reasonable adjustments. To improve the recognition of intellectual disability for adults during health service contacts, we advocate for the implementation of targeted initiatives (such as a nationwide disability flag to be included in health service records) to improve the provision of reasonable adjustments.

2) Method: Definition of the study population is missing.

Response: We note that the study population is currently defined under Method heading “Study population” on page 7. We postulate that the use of “study cohort” rather than “study population” may have been confusing. We have changed references to the cohort to “study population”, and we have changed the Study Population section, so it now reads:

The study population comprised of people with intellectual disability of ages 18 and over known to NSW disability services who appeared in the DS-MDS, PG, or SDS between 1 Jul 2005 and 30 Jun 2014, with their first appearance being the index date for the study (noting that this allowed some people to have an index date prior to the date they turned 18, as long as they turned 18 between 1 Jul 2005 and 30 Jun 2014). We excluded from the study population people with intellectual disability who did not have any hospital episodes after the index date in our study period. Persons younger than 18 years of age were excluded. 

3) Method: The authors have not mentioned the type of this study.

Response: The study is a retrospective study, we now mention this explicitly in the methods on page 6. 

4) Method: There are no details about the comparison group.

Response: This study does not involve a comparison group. All the analysis is done with the aim of predicting the presence or absence of intellectual disability within people who have intellectual disability. We hope that the changes to points 2, 3, 5, and 6 will help clarify the study design. 

5) Method: What is the outcome variable?

Response: The outcome variable was the presence or absence of an International Classification of Diseases (10th Edition) code for intellectual disability (F7, F84.2, Q90, Q91, Q93, Q95-99. Q86, Q87, Q89.8, P04.3) during an inpatient episode, which is currented detailed in the Methods section under the “Outcome measures” heading on pages 7 & 8. Again, we postulate the use of “outcome measures” rather than “outcome variable” may have been confusing, so we have altered the section for clarity with this in mind. It now reads:

Outcome variables

We measured our outcome variables at an episode of care level. The episode of care is the period of admitted patient care between a formal (cessation of a stay in hospital) or statistical (cessation of an episode occurring within a hospital stay, which may be followed by another episode) separation, characterised by only one care type.(23–25) Individuals were considered as having the first outcome variable, intellectual disability recognition, if intellectual disability was recorded in a hospital episode as a principal or additional diagnosis. We used the International Classification of Diseases 10th Edition (ICD-10) codes to identify intellectual disability (F7, F84.2, Q90, Q91, Q93, Q95-99. Q86, Q87, Q89.8, P04.3).(22)

The second outcome variable was the length of stay in days for a given hospital episode, obtained by the duration in days between episode start and end as recorded in the APDC. Episodes that started and finished within the same day were considered to have a length of stay one day. Episodes that were recorded as occurring within another episode (i.e., ‘nested’ episodes), were included as it could not be determined that these stays did not constitute discrete service contacts. 

6) Method: Was each case followed up till the outcome? & 7) Method: Details on statistical analysis are not sufficient. If we followed up the participants till the outcome, should not be the Cox regression more appropriate?

Response: All outcome variables are measured at an episode of care level, as opposed to at the person level. That is, we do not follow-up each person until they have an episode of care where they are recognised as having intellectual disability. Rather, we consider all their episodes of care after they enter the study, and examine each episode separately to see whether intellectual disability was recognised in that episode of care or not. 

We have attempted to clarify this approach under the “Outcome measures” heading in the Methods on page 7 & 8. The section now reads:

We measured our outcome variables at an episode of care level. The episode of care is the period of admitted patient care between a formal (cessation of a stay in hospital) or statistical (cessation of an episode occurring within a hospital stay, which may be followed by another episode) separation, characterised by only one care type.[23–25] That is, an episode of care can be thought of as a discrete contact with the hospital inpatient system where a person receives a particular type of care (e.g., mental health care, dialysis, rehabilitation, etc). We included all hospital episodes that occurred for an individual within the follow-up period when determining our outcome variables. Individuals were considered as having the first outcome variable, intellectual disability recognition, if intellectual disability was recorded in a hospital episode as a principal or additional diagnosis. We used the International Classification of Diseases 10th Edition (ICD-10) codes recorded for each hospital episode to identify the presence or absence of intellectual disability in a hospital episode (F7, F84.2, Q90, Q91, Q93, Q95-99. Q86, Q87, Q89.8, P04.3).[22]

8) Result: “intellectual disability was recognized in 19,261 (23.79%) of all episodes”- from what point does the author deduce that?

Response: This is drawn from the percentage of episode that had the outcome variable (one of the relevant International Classification of Diseases codes (10th edition)) mentioned in point 5. We have now included a new table (Table 3) on page 16 detailing characteristics of hospital admissions (as suggested by Editor point 2), and moved this point to the “Sociodemographics and episode characteristics” subheading on page 13 

9) Discussion: The author thoroughly detailed the result section here. The author gave no perspective or comparison.

Response: As in response to Editor point 5) we have now expanded our discussion to contrast with the paper by Bourke et al. on Intellectual Disability Recognition in Children.

10) Discussion: The authors also failed to cite many related articles

Response: We have searched again for articles pertaining to predictors of intellectual disability recognition in the health system and found only Bourke et al. to have conducted a study with direct relevance to ours. We acknowledge a small possibility that this could represent a fault in our search method, and as such welcome specific suggestions by the reviewer to literature that we may have missed. We are also happy to expand on specific topics as they relate to our findings at the discretion of the editor. 

Response: We have now correctly formatted our documents for PLOS ONE

2) Thank you for stating the following in the Acknowledgments Section of your manuscript:

“JT is funded as the Chair of Intellectual Disability Mental Health by the Mental Health Branch of the NSW Ministry of Health.”

 “This study was funded by a National Health and Medical Research Council Australia (https://www.nhmrc.gov.au/) funded Partnerships for Better Health grant (APP1056128; Title: Improving the Mental Health Outcomes of People with an Intellectual Disability; Awarded to: JT, EE, RL, LD, KF, AJ, KD) and a National Health and Medical Research Council Australia funded Project grant (APP1123033; Title: Understanding health service system needs for people with intellectual disability; Awarded to: JT, CV, NL, RM, SR). The funders had no role in study design, data collection and analysis, decision to publish, or preparation of the manuscript.”

Response: This has now been amended, our funding statement now reads:

We have amended our funding statement as follows: “This study was funded by a National Health and Medical Research Council Australia (https://www.nhmrc.gov.au/) funded Partnerships for Better Health grant (APP1056128; Title: Improving the Mental Health Outcomes of People with an Intellectual Disability; Awarded to JT) and a National Health and Medical Research Council Australia Project grant (APP1123033; Title: Understanding health service system needs for people with intellectual disability; Awarded to JT). JT is funded as the Chair of Intellectual Disability Mental Health by the Mental Health Branch of the NSW Ministry of Health. The funders had no role in study design, data collection and analysis, decision to publish, or preparation of the manuscript.”

3) In your Data Availability statement, you have not specified where the minimal data set underlying the results described in your manuscript can be found. PLOS defines a study's minimal data set as the underlying data used to reach the conclusions drawn in the manuscript and any additional data required to replicate the reported study findings in their entirety. All PLOS journals require that the minimal data set be made fully available. For more information about our data policy, please see http://journals.plos.org/plosone/s/data-availability.

Response: The nature of our data means that we cannot share the data publicly. We have amended our data statement to read:

Our datasets cannot be shared publicly due to the data usage agreement between the Department of Developmental Disability Neuropsychiatry, The University of New South Wales Sydney, and the data custodians who provide access to our data. Researchers who meet the criteria for obtaining access to confidential data who wish to access the data should enquire to the NSW Population and Health Services Research Ethics Committee (cinsw-ethics@health.nsw.gov.au) and quote the project and sub-study reference number (2013/02/446, 2019UMB0209).

Response: This has been completed

---

## [Decision Letter · Decision Letter 1]

7 Mar 2022

PONE-D-21-32313R1Predictors and outcomes of recognition of intellectual disability for adults during hospital admissions: A retrospective data linkage study in NSW, AustraliaPLOS ONE

Dear Dr. Trollor,

Thank you for submitting your manuscript to PLOS ONE. After careful consideration, we feel that it has merit but does not fully meet PLOS ONE’s publication criteria as it currently stands. Therefore, we invite you to submit a revised version of the manuscript that addresses the points raised during the review process.

We look forward to receiving your revised manuscript.

Kind regards,

Rashidul Alam Mahumud, MPH, MSc, PhD

Academic Editor

PLOS ONE

Journal Requirements:

Reviewers' comments:

Reviewer's Responses to Questions

**Comments to the Author**

1. If the authors have adequately addressed your comments raised in a previous round of review and you feel that this manuscript is now acceptable for publication, you may indicate that here to bypass the “Comments to the Author” section, enter your conflict of interest statement in the “Confidential to Editor” section, and submit your "Accept" recommendation.

Reviewer #2: (No Response)

2. Is the manuscript technically sound, and do the data support the conclusions?

Reviewer #2: Yes

3. Has the statistical analysis been performed appropriately and rigorously? 

Reviewer #2: Yes

4. Have the authors made all data underlying the findings in their manuscript fully available?

Reviewer #2: Yes

5. Is the manuscript presented in an intelligible fashion and written in standard English?

Reviewer #2: Yes

6. Review Comments to the Author

Reviewer #2: The article entitled “Predictors and outcomes of recognition of intellectual disability for adults during hospital admissions: A retrospective data linkage study in NSW, Australia” is representing the factors and outcome associated with the intellectual disability for adults during hospital admissions. The overall manuscript (MS) sounds good, well represented, and described. I think it will contribute more for the aged people related research. I would like to take attention of the authors to address/explain the following concerns for updating the MS before publication:

(i) The intellectual disability basically develops at the young age which may affect the next ages. I am not sure that the participants (especially the elders) were asked whether they were developed a confirmed ID syndrome in their early stages of life (any medical confirmation about that??). Author should clear these although they mentioned autism spectrum disorder (ASD) which is a developmental disability that can cause significant social, communication and behavioral challenges.

(ii) Another point is that the elderly people are more likely to develop Dementia/Alzheimer disease. I would like to request to add this type of description whether the participants had their diagnosis of Dementia/Alzheimer disease before?? Although the participants were asked about mental disorder.

7. PLOS authors have the option to publish the peer review history of their article (what does this mean?). If published, this will include your full peer review and any attached files.

Reviewer #2: No

---

## [Author Response · Author response to Decision Letter 1]

10 Mar 2022

1) The intellectual disability basically develops at the young age which may affect the next ages. I am not sure that the participants (especially the elders) were asked whether they were developed a confirmed ID syndrome in their early stages of life (any medical confirmation about that??). Author should clear these although they mentioned autism spectrum disorder (ASD) which is a developmental disability that can cause significant social, communication and behavioral challenges.

Answer: Our project did not involve the collection of data directly from participants through interviews – all the data used in the project was acquired via administrative datasets (which included all relevant diagnostic flags, including those for intellectual disability and autism spectrum disorder). We have attempted to clarify this point in our Methods section on page 6, where we have added the following section in italics to paragraph 1:

“This study was a retrospective cohort study, conducted as a substudy within a bigger data linkage infrastructure project described in Reppermund et al.[22] All data used in this study were collected into administrative datasets during the interaction of members of the cohort with administrative services (e.g., hospital or disability services). No participant data were gained through direct interaction with members of the cohort, such as via structured interviews. Linkage was performed by the NSW Centre for Health Record Linkage using best practice methods.” 

2) Another point is that the elderly people are more likely to develop Dementia/Alzheimer disease. I would like to request to add this type of description whether the participants had their diagnosis of Dementia/Alzheimer disease before?? Although the participants were asked about mental disorder.

Answer: We have carefully considered this point, and based on the combined experience of the team in both the study of intellectual disability and dementia, believe that the addition of a discussion around dementia would not be relevant to the current paper. Our reasoning is that dementia and intellectual disability represent different groups of disorders, and based on experience with dementia and intellectual disability in other projects, we believe dementia would not present a confounding factor to the recognition of intellectual disability in hospital settings.

---

## [Editor Report · Decision Letter 2]

14 Mar 2022

Predictors and outcomes of recognition of intellectual disability for adults during hospital admissions: A retrospective data linkage study in NSW, Australia

PONE-D-21-32313R2

Dear Dr. Trollor,

We’re pleased to inform you that your manuscript has been judged scientifically suitable for publication and will be formally accepted for publication once it meets all outstanding technical requirements.

Kind regards,

Rashidul Alam Mahumud, MPH, MSc, PhD

Academic Editor

PLOS ONE
---

## [Editor Report · Acceptance letter]

16 Mar 2022

PONE-D-21-32313R2 

Predictors and outcomes of recognition of intellectual disability for adults during hospital admissions: A retrospective data linkage study in NSW, Australia 

Dear Dr. Trollor:

I'm pleased to inform you that your manuscript has been deemed suitable for publication in PLOS ONE. Congratulations! Your manuscript is now with our production department. 

Kind regards, 

on behalf of

Dr. Rashidul Alam Mahumud 

Academic Editor

PLOS ONE